# How common are high-risk coronavirus contacts? A video-observational analysis of outdoor public place behavior during the COVID-19 pandemic

Joska Appelman[1], Lasse Suonperä Liebst[1,2], Marie Rosenkrantz Lindegaard[1,2,3]*

**1** Netherlands Institute for the Study of Crime and Law Enforcement (NSCR), Amsterdam, the Netherlands, **2** University of Copenhagen, Copenhagen, Denmark, **3** University of Amsterdam, Amsterdam, the Netherlands

* MRLindegaard@nscr.nl

## Abstract

Epidemiological evidence and recommendations from the World Health Organization suggest that close face-to-face interactions pose a particular coronavirus transmission risk. The real-life prevalence and nature of such high-risk contacts are understudied, however. Here, we video-observed high-risk contacts in outdoor public places in Amsterdam, the Netherlands, during the COVID-19 pandemic. We found that high-risk contacts were relatively uncommon: Of the 7,813 individuals observed, only 20 (0.26%) displayed high-risk contacts. Further, we qualitatively examined the 20 high-risk contacts identified and found that they occurred disproportionally between affiliated persons engaged in affiliative behaviors. We discuss the potential public health implications of the relatively low incident rate of high-risk contacts.

## Introduction

Social distancing in indoor and outdoor settings has been a critical non-pharmaceutical measurement to curb the global spread of the coronavirus. For example, in the Netherlands, public health agencies have advised maintaining 1.5 meters distance from people with whom one does not share a household [1]. However, not every behavioral violation of these social distancing directives is equally risky for coronavirus transmission. Evidence suggests that coronavirus primarily spreads via respiratory droplets during close face-to-face contact [2]. In line with this, the World Health Organization (WHO) [3] defines high-risk situations as contacts with a probable or confirmed COVID-19 case via physical contact or via face-to-face contact within one meter for at least 15 minutes.

Studies examining social distancing compliance do often not adhere to these insights and WHO's definition of high-risk contacts. This is linked to the methodological reliance on self-reported or geo-tracking measures of social distancing compliance, which are too coarse-grained to capture whether people violate the specific behavioral criteria of high-risk contacts [4, 5]. Relatedly, there is a mismatch between people's high self-reported willingness to

**Data Availability Statement:** Data, scripts, materials, and transcripts required to replicate the current results are available at osf.io/7ek9d. The raw footage cannot be shared publicly due to

conditions attached to the project approvals.
However, access to this footage may be granted
after becoming affiliated with the Netherlands
Institute for the Study of Crime and Law
Enforcement (NSCR) and signing a statement that
the applicant "will only use the data for scientific
purposes, will not make the data accessible to third
parties, and will not publish results that can
disclose the identity of the subjects on the data." To
request this access, please contact the data
manager at the NSCR, Thomas Hoogenboom
(THoogenboom@nscr.nl).

**Funding:** This study was funded by The
Netherlands Organisation for Health Research and
Development (ZonMw, www.zonmw.nl) by means
of a grant awarded to MRL (project number
10430022010017). ZonMw had no role in study
design, data collection, analysis, decision to
publish, or preparation of the manuscript.

**Competing interests:** The authors have declared
that no competing interests exist.

distance [6] and direct observational evidence showing that people often cannot distance in practice [7–9]. With re-openings of societies around the globe, it is valuable for policymakers to have precise knowledge on high-risk contacts in outdoor settings: how common are these encounters and in what situations do they occur—and should potentially be targeted?

The current study addresses these pressing public health questions using a methodological approach that offers uniquely fine-grained insight into human interpersonal behavior: video-assisted naturalistic observation [10, 11]. In doing so, we provide—to our best knowledge—the first systematic examination of the real-life prevalence and nature of high-risk contacts as defined by the WHO.

## Method

Data were footage of public places in Amsterdam, the Netherlands, recorded by three municipal surveillance cameras during the COVID-19 pandemic and provided by the Amsterdam Police Department (note that data were sourced from a wider pool of footage, which has been analyzed for different study purposes or using other methodologies [7–9, 12]). The project was approved by the Netherlands Public Prosecution Service (PaG/BJZ/49986), the Danish Data Protection Agency (514-0011/18-2000), and the Ethics Review Board of the Faculty of Social and Behavioral Science at the University of Amsterdam (2021-AISSR-14225).

The included cameras were located in relatively busy settings (i.e., shopping streets, public transportation), and the obtained footage was recorded on Thursdays and Saturdays between 9 a.m. and 8 p.m., from March 2020 to March 2021. Due to technical issues, the footage was often missing after 2 p.m.—therefore, and because video data coding is very labor-intensive, we only included footage from a short period between 1 p.m. and 1.30 p.m. This time of day was also chosen because it captured a relatively average level of pedestrian movement in public places [13]. Finally, we note that across the included months of footage, the COVID-19 infection rate fluctuated following European patterns, and a range of mitigation measures was implemented [12].

The coding was conducted by four trained research assistants and began by randomly selecting persons present on the footage—specifically, every third person crossing the streets for each camera throughout the observation period. In total, we video-observed 7,813 individuals. For each selected person, we recorded for a maximum of 120 seconds ($M = 27.1$, $SD = 15.0$, min. = 4, max. = 120) whether the person was involved in any social distancing violations. Here, we both applied a *broad*—or low-risk—definition of social distancing violations involving any interpersonal proximity within 1.5 meters [8], and a more *narrow* definition in line with the WHO's high-risk definition: face-to-face encounters between two or more people who did not arrive together, and who were within less than one meters proximity for longer than 15 minutes, or had direct physical contact. Note that we excluded people arriving together from the definition of social distancing violations, given that these persons were likely to be affiliated [14] and belong to the same household and thus are falling outside the definition of a risk encounter.

Note that the interrater reliability of the high-risk contact measure was not tested, given that we implemented this measure at a late stage in the coding and research process. However, we did evaluate this for two related social distancing measures, which both reached acceptable $AC_1$ agreement scores [15]—i.e., the broadly defined social distancing measure had an $AC_1$ of 0.97, and a version of this measure restricted to interpersonal proximities lasting a minimum of 10 seconds had an $AC_1$ of 0.84. Given that the high-risk contact measure is a subset of these measures, these acceptable scores demonstrate indirectly that the current high-risk contact measure most plausibly has reasonable reliability.

For each person involved in a high-risk contact, we furthermore conducted a detailed qualitative description of the event (e.g., type of contact and activity) and the persons involved in the encounter (e.g., their age and whether they were affiliated [14]), with the ambition of attaining a more fine-grained understanding of the behavioral sequences that led to the contacts [16]. As part of this analysis, we also categorized the types of high-risk situations according to five categories: 1) incidental touching, 2) asking questions, 3) catching-up (i.e., people meet, talk, and leave separately), 4) meeting (i.e., people meet, talk, and leave together), and 5) group reassembling (e.g., persons are waiting on each other, reassemble, and leave together).

Note that the regression and interrater reliability analyses were run with Stata 16. The coding of the video clips was done in SPSS 26.0. Replication data, scripts, and materials are available at osf.io/7ek9d.

## Results

Of the 7,813 persons observed, 6,108 or 78% (CI 95% [77%, 79%]) were within 1.5 meters of another person. For the vast majority of these common violations, the encounters remained low-risk, see Fig 1. Only 20 persons or 0.26% (CI 95% [0.14%, 0.37%]) were involved in a more narrowly defined high-risk encounter.

Note that these figures represent the rate of contacts during the average observation time of 27 seconds. To assess these rates on a more standard time unit, we regressed social distancing on observation time. Observation time (in seconds) was found to be positively associated with social distancing measured on three ordinal levels (OR = 1.05, 95% CI [1.04, 1.05], $p < .001$), i.e., no violation, 1.5 meter violation, and high-risk contact.

This ordinal regression result is graphed as predicted probabilities in Fig 2 [17]. As we see, a large proportion of persons was involved in 1.5-meter distancing violations within seconds of observation, with the predicted probability approximating 1.0 after around 1.5 minutes. This pattern is consistent with the interpretation that the 1.5-meter violation risk is high because public place crowding offers many "situational opportunities" for brief encounters [7], which, in turn, accumulates mechanically as people move through space and pass by additional people [18]. By comparison, only a small proportion of persons was involved in high-risk contact within seconds of observation, with the predicted probability only increasing noticeably after one minute. A possible explanation of this slope profile was examined as part of our qualitative assessment of the high-risk encounters.

In 14 of the 20 (70%) high-risk encounters, the individuals appeared to be affiliated rather than strangers—and within the subset of these affiliation cases, the persons were equally likely to be involved in catching up (36%), meeting (36%), and group reassembling (29%). None of

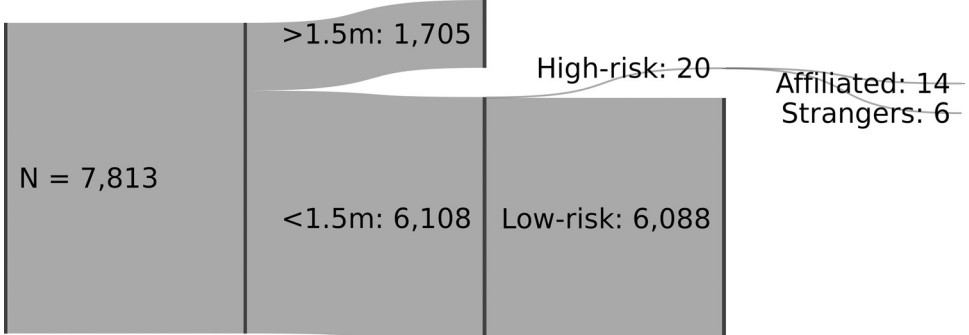

**Fig 1. Descriptive breakdown of the observed person's involvement in risk behaviors.**

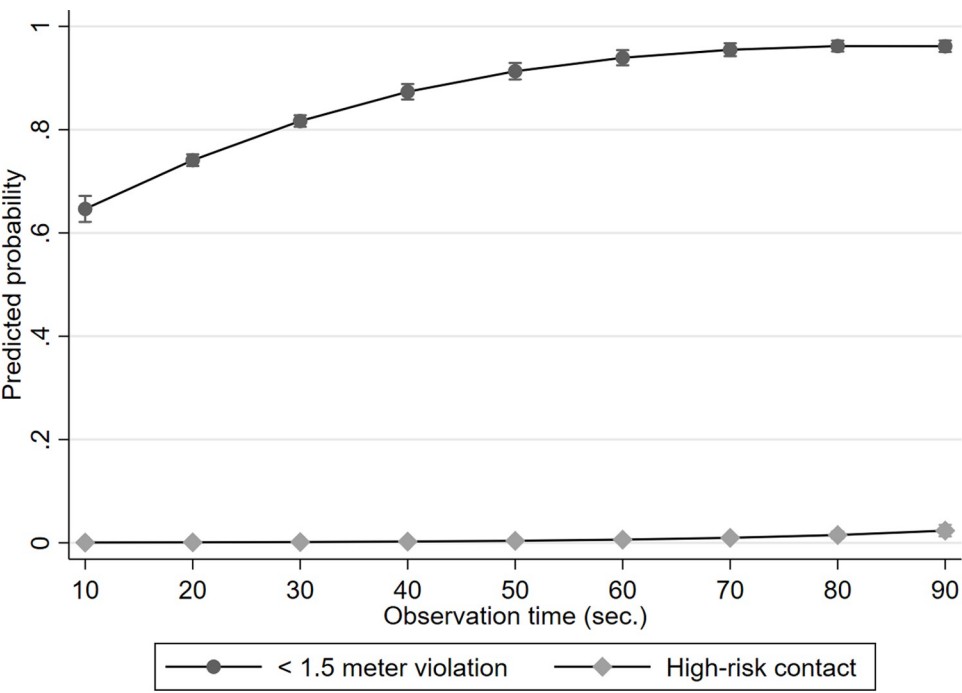

**Fig 2. 1.5-meter and high-risk violations regressed on observation time.** *Note.* The ordinal regression model was estimated with three outcome levels, but for brevity, we do not graph the no-violation outcome level.

these face-to-face activities lasted longer than fifteen minutes, but all of them involved social touching. Note that the two remaining activities (i.e., incidental touching and asking questions) did not occur between affiliated persons.

To illustrate how the interaction between affiliated persons may lead to high-risk contacts, we provide a video transcript of an event involving two persons catching up: A man walked down a shopping street with a wide walking area, shops on both sides, and a few scattered benches. He passed another man in the middle of the street and turned his head. The two men walked toward each other, shook hands, and bumped fists. They talked for a moment, bumped fists again, and continued walking in separate directions.

In the remaining six of the 20 (30%) high-risk encounters, the involved persons appeared to be strangers. Within this subset, the contacts were linked to asking questions (67%) or incidental touching (33%). Cases of asking questions involved, for example, borrowing a lighter, begging for money, or persons handing out flyers to others on the street while chatting with them. The following video transcript illustrates such a case: A man is walking along the same shopping street as in the previous example. He slows down and enters a queue in front of a pharmaceutical shop—while keeping a proper distance from the person in front of him. A young man approaches the queue, stops in front of the man, and stays within one-meter distance while saying something to him. The man reaches in his pocket and hands an object to the young man, apparently a lighter. During this interaction, the hands of both man touch. The young man lights a cigarette while remaining within a one-meter radius and returns the lighter. Then, the young man walks away from the man, who remains in the queue, and lights a cigarette himself.

The most noteworthy pattern in the qualitative analysis is that high-risk contacts appeared disproportionally associated with encounters involving affiliated persons engaged in affiliative behaviors. Adding to this interpretation, a binominal probability test offered some evidence

that person affiliation is a necessary condition for high-risk contacts [19]: The proportion of affiliated (14/20 = 70%) was significantly greater than a benchmark proportion of 50% (i.e., a null-hypothesis assuming an equal ratio of affiliated and strangers), evaluated with an alpha threshold of 0.10 (which is sensible to use given the small sample size [20]). Relatedly, the qualitative analysis shed light on the mechanism underpinning high-risk contacts. Rather than incidental encounters accumulating with movement through space (they occur, but rarely), these are mostly cases of focused interactions with some duration [21]. People meet, halt, and interact, and therefore they tend to be observed for a disproportionally longer time, as captured in Fig 1.

## Discussion

This study examined the frequency and qualitative nature of social distancing violations in public places—with a particular focus on high-risk contacts, to which prior research has shown limited attention. Our analysis demonstrated that the incident rate varied dramatically across social distancing definitions. Evaluated with a broad (and often legally sanctioned) definition that includes any interpersonal proximities within 1.5 meters, the violations were very common—i.e., similar to what is reported in prior observational studies [7–9]. However, the incident rate was comparably much lower when assessed with the WHO's [3] more narrow definition, indicating that it is possible to be present in public space with limited exposure to high-risk contacts. In a further qualitative assessment, we found that the high-risk cases were disproportionally associated with encounters between affiliated persons engaging in friendly affiliative interactions involving social touching.

The low high-risk incident rate found in the current data adds an important nuance to the view that social distancing violations are often practically impossible to avoid in crowded urban settings [7–9]. Our result is understandable from the perspective of social-behavioral research, showing that everyday encounters among strangers in urban public places are typically brief and follow a norm of noninvolvement [21, 22]. By comparison, related research demonstrates that close interactions and social touches are more common and preferred among affiliated persons than strangers [21, 23], as we also observed.

The current results have several potential implications. To mitigate high-risk encounters in public, public health agencies could focus less on stranger encounters and more on coincidental interactions between affiliated. Further, public health agencies should perhaps focus less on implementing social distancing directives in outdoor public places than has been the case throughout the COVID-19 pandemic. Not only are high-risk contacts—as defined by the WHO—relatively uncommon in this setting, but the transmission risk is also lower outdoors than indoors [24]. On the other hand, recent coronavirus mutations have enhanced the infectivity and thus increased the transmission risk in outdoor settings [25, 26]. As such, these changing virological circumstances complicate what should be behaviorally defined as a high-risk contact, and we leave it to others—scholars and public health agencies alike—to determine whether or not the low high-risk incident rate should be considered epidemiologically trivial.

We acknowledged that the current study might have limited generalizability to other national urban contexts, times of day (e.g., rush hours), and public settings (e.g., semi-public indoor places such as bars and grocery shops). For example, national cultures have varying preferences for interpersonal distance and gesturing [27, 28], and cities have rhythms of rush hour and off-peak pedestrian movement [12, 13], and such circumstances may, in turn, influence the frequency of high-risk encounters. Another study limitation concerns our strictly *behavioral* focus, without data on the actual epidemiological risks associated with the contacts we denote as "high-risk." We recommend that future research attempts to tie the behavioral

and epidemiological dimensions together in studies examining how coronavirus transmission is not merely influenced by social distancing *per se* but by how this behavior is displayed *in situ* (e.g., among affiliated, with touches). Finally, we acknowledge that the exclusion of people arriving together from the definition of social distancing may have underestimated the high-risk incident rate. As such, it should be kept in mind that the reported incident rate concerns *new* encounters rather than *any* interpersonal proximities within 1.5 meters.

## Supporting information

**S1 File. Full results of regression analysis.**
(TXT)

## Acknowledgments

We thank Josephine Thomas, Kiki Bijleveld, and Laura Hendriks for their involvement in the coding of the data and the Amsterdam Police Department for facilitating the data collection.

## Author Contributions

**Conceptualization:** Lasse Suonperä Liebst, Marie Rosenkrantz Lindegaard.

**Data curation:** Joska Appelman, Lasse Suonperä Liebst.

**Formal analysis:** Joska Appelman, Lasse Suonperä Liebst.

**Funding acquisition:** Marie Rosenkrantz Lindegaard.

**Investigation:** Joska Appelman.

**Methodology:** Joska Appelman, Lasse Suonperä Liebst, Marie Rosenkrantz Lindegaard.

**Project administration:** Marie Rosenkrantz Lindegaard.

**Resources:** Marie Rosenkrantz Lindegaard.

**Supervision:** Marie Rosenkrantz Lindegaard.

**Writing – original draft:** Joska Appelman.

**Writing – review & editing:** Lasse Suonperä Liebst, Marie Rosenkrantz Lindegaard.

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
