## [Decision Letter · Decision Letter 0]

10 Dec 2021

PONE-D-21-34926How common are high-risk coronavirus contacts? A video-observational analysis of outdoor public place behavior during the COVID-19 pandemicPLOS ONE

Dear Dr. Appelman,

Thank you for submitting your manuscript to PLOS ONE. After careful consideration, we feel that it has merit but does not fully meet PLOS ONE’s publication criteria as it currently stands. Therefore, we invite you to submit a revised version of the manuscript that addresses the points raised during the review process.

We look forward to receiving your revised manuscript.

Kind regards,

Sanjay Kumar Singh Patel, Ph.D.

Academic Editor

PLOS ONE

Journal Requirements:

Reviewers' comments:

Reviewer's Responses to Questions

**Comments to the Author**

1. Is the manuscript technically sound, and do the data support the conclusions?

Reviewer #1: Yes

Reviewer #2: Partly

Reviewer #3: Yes

2. Has the statistical analysis been performed appropriately and rigorously? 

Reviewer #1: Yes

Reviewer #2: Yes

Reviewer #3: Yes

3. Have the authors made all data underlying the findings in their manuscript fully available?

Reviewer #1: Yes

Reviewer #2: No

Reviewer #3: Yes

4. Is the manuscript presented in an intelligible fashion and written in standard English?

Reviewer #1: Yes

Reviewer #2: Yes

Reviewer #3: Yes

5. Review Comments to the Author

Reviewer #1: This very research article entitled " How common are high-risk coronavirus contacts? A video-observational analysis of outdoor public place behavior during the COVID-19 pandemic" by Appelman et al., video-observed high-risk contacts in outdoor public places in Amsterdam, the Netherlands, during the COVID-19 pandemic. Authors found that high-risk contacts were relatively uncommon as out of 7,814 individuals observed, only 20 (0.26%) displayed such contact. While the topic is of increasing relevance, still, this reviewer has certain suggestions that would help produce a more comprehensive study of the topic:

Specific comments that the authors should consider

1, Authors have performed this study at outdoor public places in Amsterdam, it would be interesting to analyze other available data from other geographical location in the Netherlands.

2, Authors should provide their data graphical representable format.

3, Authors should discuss their study by putting/citing some current research.

4, At least one supplementary Figure as illustration may be afforded as to highlight the summary or prospect of this study.

5, What was the infection rate in Netherlands when this study was conducted and what might be the proposed reason for that?

Reviewer #2: This paper uses video-observational analysis to investigate the high-risk contact behaviour at outdoor public place during the COVID-19 pandemic. According to WHO, high-risk situation is defined as contacts with a COVID-19 patient via physical contact or via face-to-face contact within one meter, for at least 15 minutes. And the authors find that such high-risk contacts were relatively uncommon. The authors thus conclude that their results alleviate the public health concern regarding coronavirus transmission in outdoor public places.

This conclusion may mislead. From the several waves of SARS-CoV-2 virus infection outbreak in the past two years in almost all major countries in the world, we can see that although the rate of outdoor person-to-person viral transmission is much lower than that of the indoor transmission, outdoor transmission is still a considerable form of transmission. The authors’ result rather shows that the WHO definition of high-risk contact is inadequate: long distance (greater than one meter) and short time (less than 15 minutes) can still constitute high risk of infection. This is because, the SARS-CoV-2 virus can be transmitted in airborne way [1-3].

The highly contagious nature of the SARS-CoV-2 viral infection shows that although facial mask may slow down the transmission of the virus, in long run, infection is inevitable, and we have to live with this SARS-CoV-2 virus. Epidemiologic data show that most of the COVID-19 cases are asymptomatic and mild [4] and the SARS-CoV-2 virus is self-limiting [5]. The severe cases of COVID-19 is dominantly caused by the immunopathology, i.e., the overreaction of the host immune system and the cytokine storm [6,7] instead of viral infection. So the public health authorities may focus more on the curing of the noncommunicable diseases like morbid obesity, type 2 diabetes mellitus and other metabolic syndromes than the concern regarding the coronavirus infection and transmission.

Reference

1. Huang J, Jones P, Zhang A, Hou SS, Hang J and Spengler JD (2021) Outdoor Airborne Transmission of Coronavirus Among Apartments in High-Density Cities. Front. Built Environ. 7:666923. doi: 10.3389/fbuil.2021.666923

2. Yu ITS, Li Y, Wong TW (2004) Evidence of airborne transmission of the severe acute respiratory syndrome virus. N. Engl. J. Med. 350(17):1731–1739. DOI: 10.1056/NEJMoa032867.

3. Zhang R, Li Y, Zhang AL, Wang Y, Molina MJ (2020) Identifying airborne transmission as the dominant route for the spread of COVID-19. Proc Natl Acad Sci U S A. 117(26):14857-14863. DOI: 10.1073/pnas.2009637117.

4. Wu Z, McGoogan JM (2020) Characteristics of and Important Lessons From the Coronavirus Disease 2019 (COVID-19) Outbreak in China: Summary of a Report of 72 314 Cases From the Chinese Center for Disease Control and Prevention. JAMA. 323(13):1239-1242. DOI: 10.1001/jama.2020.2648

5. Zhu CC, Zhu J (2021) The effect of self-limiting on the prevention and control of the diffuse COVID-19 epidemic with delayed and temporal-spatial heterogeneous. BMC Infect Dis 21, 1145. DOI: 10.1186/s12879-021-06670-y

6. Cao, X (2020) COVID-19: immunopathology and its implications for therapy. Nat Rev Immunol 20, 269–270. DOI: 10.1038/s41577-020-0308-3

7. van Eijk LE, Binkhorst M, Bourgonje AR, Offringa AK, Mulder DJ, Bos EM, Kolundzic N, Abdulle AE, van der Voort PH, Olde Rikkert MG, van der Hoeven JG, den Dunnen WF, Hillebrands JL, van Goor H (2021) COVID-19: immunopathology, pathophysiological mechanisms, and treatment options. J Pathol. 254(4):307-331. DOI: 10.1002/path.5642.

Reviewer #3: In this paper entitled " How common are high-risk coronavirus contacts? A video-observational analysis of outdoor public place behavior during the COVID-19 pandemic", the authors used video-observed high-risk contacts in outdoor public places. The study has 7814 individuals and only 0.26 % of individuals display high-risk contact. The manuscript alleviates public health concerns regarding coronavirus transmission in outdoor places based on a low number of high-risk individuals. In addition, the manuscript is fascinating and easy to understand. However, there are a few concerns in the manuscript.

Minor Comments:

1) The English may be polished. There are grammatical errors and spelling mistakes in the manuscript.

2) The link provided in the manuscript in line 100 is not working. Find a way to make the data available to readers.

3) The cameras were located in busy settings such as shopping streets, public transportation. Will footage from pubs, grocery shops, coffee shops change the results?. Why did authors not consider these settings for their study as these sites are closer and human interaction is more?.

4) The footage is from 1 p.m and 1.30 p.m; for half an hour in the afternoon. During this time, the rush in public transport is minor. Why did the authors not consider another time for their study?. Will footage time like office hours and school hours affect the results?

5) Please mention the limitation of the study?

6) The authors may additionally provide one Figure as a summary in results section.

7) The author may elaborate on the results of observational video analysis in the discussion.

9) Discuss similar studies in the introduction and few information on social distancing, nature biomolecules for their treatment, and importance of health (few references i.e. doi: 10.1007/s12088-020-00908-0; doi: 10.1007/s12088-020-00893-4).

---

## [Author Response · Author response to Decision Letter 0]

28 Jan 2022

*** Response to reviews of ‘How common are high-risk COVID-19 contacts?’****

***Editors feedback***

***REPLY: Validated. 

***REPLY: Replication data and materials are available at https://osf.io/7ek9d/.

***REPLY: We would like to stress that all data scripts, transcripts, and materials are publicly available at https://osf.io/7ek9d/. Only the raw data, consisting of video recordings of individuals moving through public space, is available upon request. Ethical and legal restrictions for sharing this data are mentioned in the revised Data Availability statement.

***Reviewer #1***

4. Authors have performed this study at outdoor public places in Amsterdam, it would be interesting to analyze other available data from other geographical location in the Netherlands.

***REPLY: We agree that this would be interesting and relevant. However, in the current study, we only have access to CCTV camera footage from Amsterdam. In fact, we were only able to collect this data because of our long-term collaboration with the Amsterdam Police Department—there exists no centralized, nationwide archive of CCTV data. In the revised paper (p. 9), we have further flagged this geographical issue as a study limitation. 

5. Authors should provide their data graphical representable format.

***REPLY: We agree that this would add clarity to the argument and have thus summarized the data patterns in Figure 1 (p. 5) and Figure 2 (p. 6). 

6. Authors should discuss their study by putting/citing some current research.

***REPLY: We now cite and discuss other recent studies relating to the topic at hand, both in the introduction (p. 3) and the discussion section (p. 8-9). 

7. At least one supplementary Figure as illustration may be afforded as to highlight the summary or prospect of this study.

***REPLY: In the exercise of trying to visualize our main results, we decided to create two figures. Figure 1 (p. 5) is a descriptive figure which summarizes the main findings with regard to the frequency of high-risk contacts, and Figure 2 (p. 6) is modelling based and details the difference between low and high risk distancing contacts. We have revised the Results section as part of the inclusion of these figures.

8. What was the infection rate in Netherlands when this study was conducted and what might be the proposed reason for that? 

***REPLY: In the revised paper (p. 4), we now specify that the Netherlands case largely followed the European pattern, with fluctuating infection rates across the data collections period lasting one full year. 

***Reviewer #2:***

9. The authors thus conclude that their results alleviate the public health concern regarding coronavirus transmission in outdoor public places.

This conclusion may mislead. From the several waves of SARS-CoV-2 virus infection outbreak in the past two years in almost all major countries in the world, we can see that although the rate of outdoor person-to-person viral transmission is much lower than that of the indoor transmission, outdoor transmission is still a considerable form of transmission. The authors’ result rather shows that the WHO definition of high-risk contact is inadequate: long distance (greater than one meter) and short time (less than 15 minutes) can still constitute high risk of infection. This is because, the SARS-CoV-2 virus can be transmitted in airborne way [1-3].

The highly contagious nature of the SARS-CoV-2 viral infection shows that although facial mask may slow down the transmission of the virus, in long run, infection is inevitable, and we have to live with this SARS-CoV-2 virus. Epidemiologic data show that most of the COVID-19 cases are asymptomatic and mild [4] and the SARS-CoV-2 virus is self-limiting [5]. The severe cases of COVID-19 is dominantly caused by the immunopathology, i.e., the overreaction of the host immune system and the cytokine storm [6,7] instead of viral infection. So the public health authorities may focus more on the curing of the noncommunicable diseases like morbid obesity, type 2 diabetes mellitus and other metabolic syndromes than the concern regarding the coronavirus infection and transmission.

***REPLY: We agree that the implications of our results could be nuanced along the suggested lines. The revised paper (p. 8-9) now unpacks these potential implications and considerations in a paragraph. 

***Reviewer #3:***

10. The English may be polished. There are grammatical errors and spelling mistakes in the manuscript.

***REPLY: We have copy-edited the manuscript. 

11. The link provided in the manuscript in line 100 is not working. Find a way to make the data available to readers.

***REPLY: We thank reviewer #3 for flagging this; the link is now updated. 

12. The cameras were located in busy settings such as shopping streets, public transportation. Will footage from pubs, grocery shops, coffee shops change the results?. Why did authors not consider these settings for their study as these sites are closer and human interaction is more?.

***REPLY. We agree that this would be interesting, but the CCTV data systems we have obtained access to are only installed in outdoor public places. However, in the revised paper (p. 9), we now flag this as a study limitation. Further, in the discussion (p. 9), we now also consider the current result with respect to interaction in indoor vis-à-vis outdoor settings. 

13. The footage is from 1 p.m and 1.30 p.m; for half an hour in the afternoon. During this time, the rush in public transport is minor. Why did the authors not consider another time for their study?. Will footage time like office hours and school hours affect the results?

***REPLY: We thank reviewer #3 for this comment, we now add a comment on why this period was selected, and furthermore, we acknowledge this as a study limitation that may have influenced the current results (p. 9). 

14. Please mention the limitation of the study?

***REPLY: We have further detailed the study limitations (presented in the last paragraphs of the paper (p. 9). 

15. The authors may additionally provide one Figure as a summary in results section.

***REPLY: In the exercise of trying to visualize our main results, we decided to create 2 figures. Figure 1 (p. 5) is a descriptive figure which summarizes the main findings with regard to the frequency of high-risk contacts, and Figure 2 (p. 6) is modelling based and details the difference between low and high risk distancing contacts. We have revised the Results section as part of the inclusion of these figures.

16. The author may elaborate on the results of observational video analysis in the discussion.

***REPLY: Thanks to this comment, the revised paper now elaborates/discusses the results of the analysis in more detail (p. 8). Relatedly, we have also expanded the introduction with citations and considerations of other research so as to clarify the contribution of the paper (p. 2-3). 

17. The authors discuss similar studies in the introduction and few information on social distancing, nature biomolecules for their treatment, and importance of health (few references i.e. doi: 10.1007/s12088-020-00908-0; doi: 10.1007/s12088-020-00893-4).

***REPLY: Throughout the paper revision, we have considered additional relevant citations and have included several. We hope that this helps position our contribution more clearly within the emerging social distancing and related Covid-19 literature.

---

## [Decision Letter · Decision Letter 1]

7 Mar 2022

How common are high-risk coronavirus contacts? A video-observational analysis of outdoor public place behavior during the COVID-19 pandemic

PONE-D-21-34926R1

Dear Dr. Appelman,

We’re pleased to inform you that your manuscript has been judged scientifically suitable for publication and will be formally accepted for publication once it meets all outstanding technical requirements.

Kind regards,

Sanjay Kumar Singh Patel, Ph.D.

Academic Editor

PLOS ONE

Reviewer #1: The manuscript entitled "How common are high-risk coronavirus contacts? A video-observational analysis of outdoor public place behavior during the COVID-19 pandemic " has been improved form its previous draft.

Reviewer #2: As the authors have addressed all my previous review comments, I have no further comments to the manuscript, and the manuscript can now be accepted for publication in its current state.

---

## [Editor Report · Acceptance letter]

9 Mar 2022

PONE-D-21-34926R1 

How common are high-risk coronavirus contacts? A video-observational analysis of outdoor public place behavior during the COVID-19 pandemic 

Dear Dr. Appelman:

I'm pleased to inform you that your manuscript has been deemed suitable for publication in PLOS ONE. Congratulations! Your manuscript is now with our production department. 

Kind regards, 

on behalf of

Dr. Sanjay Kumar Singh Patel 

Academic Editor

PLOS ONE